# Vaccine Hesitancy: COVID-19 and Influenza Vaccine Willingness among Parents in Wuxi, China—A Cross-Sectional Study

**DOI:** 10.3390/vaccines9040342

**Published:** 2021-04-01

**Authors:** Qiang Wang, Shixin Xiu, Shuangyu Zhao, Jianli Wang, Ying Han, Shuheng Dong, Jinxin Huang, Tingting Cui, Liuqing Yang, Naiyang Shi, Minqi Liu, Yue Han, Xuwen Wang, Yuan Shen, Enpin Chen, Bing Lu, Hui Jin, Leesa Lin

**Affiliations:** 1Department of Epidemiology and Health Statistics, School of Public Health, Southeast University, Nanjing 210009, China; 230198325@seu.edu.cn (Q.W.); 213181074@seu.edu.cn (S.Z.); 213171070@seu.edu.cn (J.W.); 220193551@seu.edu.cn (Y.H.); 213172733@seu.edu.cn (S.D.); 220183468@seu.edu.cn (J.H.); 230189704@seu.edu.cn (T.C.); 230208397@seu.edu.cn (L.Y.); 220183431@seu.edu.cn (N.S.); 220193555@seu.edu.cn (M.L.); 220193593@seu.edu.cn (Y.H.); 2Key Laboratory of Environmental Medicine Engineering, School of Public Health, Southeast University, Ministry of Education, Nanjing 210009, China; 3Wuxi Center for Disease Control and Prevention, Wuxi 214023, China; wxcdcsx@163.com (S.X.); wxmgwang@163.com (X.W.); wxcdcshy@163.com (Y.S.); chenepxia@163.com (E.C.); Lub19661023@163.com (B.L.); 4Department of Infectious Disease Epidemiology, London School of Hygiene and Tropical Medicine, Keppel Street, London WC1E 7TH, UK; Leesa.Lin@lshtm.ac.uk

**Keywords:** parents, vaccine hesitancy, COVID-19 vaccine, influenza vaccine, willingness

## Abstract

Objectives: We aimed to (1) assess parental hesitancy about category A (Expanded Program on Immunization (EPI)) and B (non-EPI) vaccines, (2) assess parental willingness for COVID-19 and influenza vaccinations, and (3) explore the association of vaccination hesitancy of parents and healthcare workers (HCWs). Methods: The study was performed in Wuxi, eastern China between 21 September 2020 and 17 October 2020. Parents of children aged <18 years and HCWs were recruited from the selected immunization clinics. Vaccine hesitancy was assessed using the Strategic Advisory Group of Experts (SAGE) vaccine hesitancy survey (VHS) by summing the total score for 10 items (maximum 50 points). Results: A total of 3009 parents and 86 HCWs were included in the analysis. The category A VHS scores were significantly higher than the category B VHS scores (*p* = 0.000). Overall, 59.3% and 52.4% of parents reported willingness to avail COVID-19 and influenza vaccination for their children, respectively; 51.2% of the HCWs wanted to be vaccinated against COVID-19. Parental category B VHS scores were associated with HCW category B VHS scores (*r* = 0.928, *p* = 0.008). Conclusions: In China, parents are more hesitant about category B vaccines than category A vaccines. More than 40% of parents showed hesitancy and a refusal to use COVID-19 and influenza vaccines.

## 1. Introduction

Vaccine hesitancy was named one of the top 10 issues threatening global health in 2019 [1]. Vaccine hesitancy is defined by World Health Organization Strategic Advisory Group of Experts (SAGE) as the delay in acceptance or refusal of vaccination despite the availability of vaccination services [2].

Children are important targets of routine vaccinations, and parental vaccine hesitancy results in a decrease in vaccine uptake among children. This has in turn led to the surge of many vaccine-preventable diseases, such as measles [3]. In 2018, the largest measles outbreak occurred in New York City, USA in nearly 30 years [4]. In European Union countries, 148,279 measles cases were reported between 2010 and 2019 [5].

Influenza-vaccine uptake among children attracts the attention of health authorities and researchers. More parents (25.8%) showed hesitancy about the influenza vaccine compared with other routine childhood vaccines (6.1%) [6]. The psychosocial factors (including necessity and safety of a vaccination for a child, etc.) contributed to parental hesitancy on influenza vaccines [7,8,9]. However, the influenza willingness rate has been reported to increase during the COVID-19 pandemic [10,11].

The current coronavirus disease 2019 (COVID-19) pandemic has caused more than 2.14 million deaths [12], and vaccination is key to controlling it. In a global survey, 54.85–88.62% of the respondents showed willingness to receive the COVID-19 vaccine [13]. Among parents, an international survey of six countries showed that 65% of parents were willing to vaccinate their child [14]. An estimated 60–72% coverage is needed to reach the herd-immunity threshold to control the spread of COVID-19 [15]. Vaccination for children may be an important part of achieving herd immunity.

In China, vaccine hesitancy needs attention: 24.9% of parents showed hesitancy to get all the recommended vaccinations for their child(ren) [16]. Meanwhile, the influenza vaccine is not regarded as important. A meta-analysis of 126 studies showed that influenza-vaccine coverage among children was lower than 30% in mainland China [17]. Approximately 20% of participants surveyed in China showed hesitancy toward the COVID-19 vaccination [13,18].

However, some issues remain unclear. Vaccination programs in China are divided into the Expanded Program on Immunization (EPI) and the non-Expanded Program on Immunization (non-EPI) [19]. Routine vaccines introduced in the EPI are referred to as category A and are free and mandatory. Meanwhile, non-EPI vaccines are referred to as category B and are optional and self-pay. Parental vaccine hesitancy might be distinctly different for categories A and B vaccines. Further, parents’ willingness regarding the COVID-19 and influenza vaccinations (category B vaccine) and associated factors were not clear in China. Obtaining data on Chinese parents’ intentions for the COVID-19 and influenza vaccines could help inform the development of child immunization policies to effectively address COVID-19 and influenza vaccine hesitancy and improve uptake in China.

The previous studies suggested that recommendations from healthcare workers (HCWs) were influencing factors of vaccine uptake among the general population [17,20,21]. HCWs working in immunization clinics have direct contact with parents, and HCWs’ attitudes and beliefs on vaccination play a role in parents’ immunization decisions [20]. However, few studies have examined the relationship between HCWs in immunization clinics and vaccine hesitancy among parents.

Thus, this study aimed to (1) assess and compare the level of hesitancy between category A and B vaccines, and the sociodemographic factors associated with vaccine hesitancy, in Chinese parents; (2) assess the parents’ willingness to acquire COVID-19 and influenza vaccines for their children, and the sociodemographic factors associated with vaccination willingness; (3) examine the association between routine vaccine hesitancy and COVID-19 vaccine willingness; and (4) explore the association between vaccine hesitancy among parents and HCWs in immunization clinics.

## 2. Methods

### 2.1. Study Design and Participants

This cross-sectional study was performed in Wuxi City, eastern China (total population: 6.59 million in 2018) between 21 September 2020 and 17 October 2020 (Appendix A). We recruited participants from local community health service center immunization clinics. The immunization clinics provide vaccination services for children and adults, and the main service populations are children aged <6 years. The participants were fathers or mothers of children aged <18 years. Only those who have lived in Wuxi City for more than 3 years were eligible. If both the father and mother came to the vaccination clinic, the parent who self-identified as the primary caregiver for the child completed the questionnaire.

The vaccination clinic was selected using a stratified sampling method. First, areas and counties in Wuxi City (Appendix A) were divided into three levels based on economic characteristics in 2019 (gross domestic product) [22]. One area was selected from each level, and two clinics in each area were selected using a random number. The HCWs in all six sampled immunization clinics were all included in the study.

The sample size was calculated using the following formula:$$N_{min}=\mathrm{deff}\times \frac{Z^{2}{}_{\left(1-\alpha \right)/2}\times p\times \left(1-p\right)}{d^{2}}$$

The type I error (α) was 0.05, the precision (*d*) was 0.04, and the design effect (deff) was 2 [16,23]. A 50% vaccine hesitancy rate (to get the largest sample size) was used because this number was not given in the previous study. The minimum target sample size was calculated to be 2081.

### 2.2. Measures

The parental questionnaire consisted of three parts: sociodemographic information, the vaccine-hesitancy scale (VHS), and COVID-19 and influenza vaccine willingness. The sociodemographic portion included information for children (date of birth, gender, firstborn status) and parents (relationship with child, date of birth, educational level, annual household income, and healthcare occupation). We divided educational level into four groups: junior high school or below, high school graduate or equivalent, college or equivalent, and master’s diploma or above. Annual household income was divided into four groups with reference to the per capita disposable income (RMB 54,847; USD 1 = RMB 6.8148) in Wuxi in 2019 [24]: <RMB 50,000; RMB 50,000–<100,000; RMB 100,000–<150,000; and ≥RMB 150,000.

Vaccine hesitancy about category A and B vaccines was assessed on a 5-point Likert scale using the SAGE 10-item VHS (Table 1) [25]. The 10-item VHS has acceptable reliability and validity, and has been used in numerous countries previously [6,26,27,28]. Each item was answered as strongly disagree, disagree, neither agree nor disagree, agree, or strongly agree. In the questionnaire, we replaced “category A” with “free”, and “category B” with “self-pay” because of the common names used by parents, in order to improve comprehension of the questionnaire.

COVID-19 and influenza vaccine willingness were measured using two questions: “If COVID-19 vaccine is available, will you vaccinate for your child?” and “Will you vaccinate your child against influenza this year?” The answers available for the two questions were “yes”, “not sure”, and “no.” Two follow-up questions on COVID-19 vaccine willingness were added: “If you choose ‘yes,’ why?” and “If you choose ‘not sure’ or ‘no,’ why?”. The participants who reported willingness to receive a COVID-19 vaccination for their child answered another two questions: “Do you want your child to be the first population to receive the COVID-19 vaccine” (yes/no/not sure) and “Will you choose domestic or foreign COVID-19 vaccine for your child?” (domestic/foreign/not sure).

Responses to the VHS questionnaire were assigned scores as follows: strongly disagree, 1 point; disagree, 2 points; neither agree nor disagree, 3 points; agree, 4 points; and strongly agree, 5 points. Responses to items L5, L9, and L10 were flipped because these three questions were worded negatively. The total score for all 10 items was summed, and total maximum score was 50 points. Lower scores represented higher vaccine hesitancy. We used the cutoff of total scores ≤30 to define “high hesitant” because this value indicated hesitancy level was the midpoint of scale scores [6]. We also set the cutoff value to ≤40 to perform the sensitive analysis.

The questionnaire for the HCWs also consisted of three parts: sociodemographic, VHS, and COVID-19 vaccine willingness. The sociodemographic portion included questions regarding gender, educational level, specialty, title, working years, and annual household income. Meanwhile, VHS and COVID-19 vaccine willingness were similar to the parents’ questionnaires.

## 3. Quality Control and Statistical Analysis

Data from the questionnaires were entered into EpiData 3.1 software, and all data were entered twice. Questionnaires with at least three missing answers were excluded. The missing data were filled using multiple imputation methods [29]. The methods and results validation of questionnaire are shown in Appendix A: Method. The results suggested that the reliability and validity of the 10-item VHS were acceptable.

Data were reported using descriptive statistics. Normally and non-normally distributed continuous variables were analyzed using the t-test and the Mann–Whitney U test, respectively. Meanwhile, categorical variables were compared using the Chi-square or Fisher’s exact test. The Spearman correlation was used to describe the relationship between VHS scores and vaccination willingness. Meanwhile, we reported the relationship between VHS scores and vaccination willingness between HCWs and parents divided by immunization clinics (using mean values).

The (*β*) coefficient and its 95% confidence interval (CI) of each variable were estimated using a generalized linear model with a maximum likelihood method to identify the influencing factors of vaccine hesitancy [30,31]. The odds ratio (OR) and 95% CI were used to explore the influencing factors of COVID-19 and influenza willingness, using logistic regression analysis. In this analysis, we combined the responses “no” and “not sure” into “no” because they both indicated a lack of confidence. Significant variables in the univariate analysis were included in the multivariate analysis. The results of significant variables were reported in the multivariate analysis. The false discovery rate (FDR) method was used to adjust the *p* value to control the error rate among significant results [32]. All statistical analyses were performed using R software. A two-sided *p* value of <0.05 was considered statistically significant.

## 4. Results

### 4.1. Sociodemographic Characteristics of the Participants

Of the 3079 questionnaires collected, 3009 questionnaires were included in the analysis (Figure 1). The average participant age was 31.36 years old, and 74.6% were mothers (Table 2). Overall, 69.7% had a college degree or above, and 93.8% did not have a healthcare occupation.

Meanwhile, 86 HCWs with an average age of 36.02 years completed the questionnaire. A total of 76.7% of the HCWs were female, and more than 50% had >10 years of working experience.

### 4.2. Parental Vaccine Hesitancy

The mean ± SD category A VHS scores were significantly higher than the category B VHS scores (40.96 ± 3.87 vs. 38.58 ± 5.05, *p* = 0.000). For items L1–L4 and L6–L8, parents showed more agreement on category A vaccines than category B vaccines (Figure 2). Overall, 0.5% and 6.8% of parents were “high hesitant” about category A and B vaccines, respectively, when the cutoff was ≤30. When the cutoff was ≤40, 46.5% and 64.5% of parents were “high hesitant” about category A and B vaccines, respectively. The fathers’ scores were also significantly higher than the mothers’ scores for category A and B VHS (Appendix A).

The multivariate analysis (Table 3) showed that compared with mothers, fathers showed less hesitancy on category A (*β*: 0.011, 95% CI: 0.003, 0.019, *p* = 0.012) and B vaccines (*β*: 0.013, 95% CI: 0.002, 0.024, *p* = 0.023). Compared with parents having annual household income < RMB 50,000, parents having annual household income RMB 50,000–< 10,000 showed less hesitancy on category A vaccines (*β*: 0.008, 95% CI: 0.000, 0.016, *p* = 0.042).

## 5. COVID-19 and Influenza Vaccination Willingness among Parents

Overall, 59.3% of parents expressed willingness to obtain the COVID-19 vaccination for their child(ren) (Appendix A). The most frequent reason (80.50%) for willing to be vaccinated was “protecting the people around”, while the most frequent reason (84.80%) for unwillingness to be vaccinated was “concern about side effects” (Appendix A). For influenza vaccination, 52.4% of the parents expressed willingness for their child to be vaccinated. Of the 3009 participants, 1109 (68.8%) were willing to take both vaccines, and 714 (51.1%) were not willing to take both vaccines (Appendix A).

In the multivariate analysis (Table 4), COVID-19 vaccine willingness was associated with educational level. Parents with a college education or below were more likely to accept COVID-19 vaccines than parents with master’s diploma or above. Influenza vaccine willingness was associated with the child’s age and annual household income. Compared with parents with an annual household income of ≥RMB 150,000, parents with annual household incomes <RMB 150,000 were less likely to accept the influenza vaccine.

## 6. The Relationship between Vaccine Hesitancy and Vaccination Willingness

The correlation between category A VHS scores and category B VHS scores COVID-19 was significant (*r* = 0.518, *p* = 0.000). There was no significant correlation between category A VHS scores and parental willingness to obtain the COVID-19 (*r* = −0.007, *p* = 0.689) and influenza (*r* = −0.001, *p* = 0.947) vaccines (Appendix A) for their child(ren). Meanwhile, the category B VHS scores were associated with parental willingness to receive the COVID-19 (*r* = 0.074, *p* = 0.000) and influenza (*r* = 0.157, *p* = 0.000) vaccine (Appendix A).

## 7. Vaccine Hesitancy and Willingness among Healthcare Workers

Among HCWs, category A VHS scores were significantly higher than category B VHS scores (43.48 ± 4.34 vs. 42.65 ± 3.89, *p* = 0.000). HCWs showed more agreement on item L1–L4, and L6–L8, compared with parents (Figure 2 and Appendix A). In total, 51.2% of the HCWs wanted to be vaccinated against COVID-19 (Appendix A). The most frequent reason (86.36%) for willing to be vaccinated was “protecting the people around”, while the most frequent reason (80.95%) for unwillingness to be vaccinated was “concern about side effects” (Appendix A).

The category A VHS scores and COVID-19 vaccination willingness among parents were not significantly correlated with those among HCWs. In contrast, there was a significant positive correlation between parents and HCWs for category B VHS scores (Figure 3).

## 8. Discussion

This study found that parents were far more hesitant about category B vaccines than category A vaccines in China. The parents showed more confidence toward category A vaccines, as evidenced by the positive responses to items L1–L4 and L6–L8 in the VHS [27,28]. More than 50% of parents expressed willingness for the COVID-19 and influenza vaccinations for their child(ren); more than 50% of HCWs expressed willingness for the COVID-19 vaccination. There was a significant positive correlation between parents and HCWs for category B VHS scores.

A previous study showed that the implementation of mandatory infant vaccinations is positively associated with the proportion of mothers in favor of vaccinations [33]. In China, parents are more confident in category A vaccines because of their trust in government-mandated vaccines [13]. In this study, more than 20% of the parents agreed that their “child/children does or do not need category B vaccines for diseases that are not common anymore”. Meanwhile, more than 20% of the parents also showed hesitancy about this item. Optional vaccines might make parents consider that a category B vaccine is unnecessary. Future vaccine communication programs need to focus on improving the parents’ confidence in category B vaccines. More qualitative studies should be carried out to explore parents’ attitudes toward category B vaccines.

Our study demonstrated that mothers were more hesitant about routine childhood vaccinations than fathers, consistent with previous studies [26,34]. A previous study suggested that men engage in riskier behaviors than do women [14]. A systematic review showed that women were less likely to be vaccinated during the 2009 global influenza pandemic [35]. Moreover, parents of first-born children were more hesitant about category A vaccines, consistent with previous studies [34,36]. For a firstborn child’s parents, less experience with childhood vaccination might contribute to more vaccine hesitancy [37]. More education for first-time parents, especially mothers, should be implemented to address parental hesitancy on routine vaccinations.

Approximately 40% of the parents were unwilling to vaccinate their children for COVID-19. Our results were lower than the data reported in the global survey (65%) [13]. The frequent reason for being unwilling to be vaccinated was “concern about side effects”. Our study showed that educational level was an effective predictor associated with COVID-19 vaccine willingness, whereas annual household income was an effective predictor associated with influenza vaccine willingness.

Government and health authorities should build a transparent, robust, and reasonable immunization process for COVID-19 vaccines with context-tailored vaccine communication that addresses public concerns. Behavioral-change theories (e.g., the health-belief model and social marketing [38,39]), which have been effectively adapted to improve individual medical use, should be used in developing strategies to improve the acceptance and uptake of COVID-19 vaccinations. Effective communication for COVID-19 vaccines should target those with graduate degrees.

The stratified analyses demonstrated that compared with parents of children aged under 3 years, the parents of children over 3 years were more willing to have their child vaccinated for influenza. However, the average annual incidence of influenza-associated death is the highest among children aged 6–23 months, excluding children aged <6 months who cannot receive the influenza vaccine [40]. These findings underline that the uptake of influenza vaccines for children aged <3 years in China needs more attention.

Consistent with a previous study, HCWs’ hesitancy toward vaccines was concerning [41,42]. HCWs showed a lack of key vaccine knowledge and less confidence in vaccines [42,43,44]. Approximately 50% of HCWs showed hesitancy toward COVID-19 vaccination, which remained a concern. The willingness data in our study was lower than 57.5% in the USA [45]; 71.6% in France, Belgium and Canada [46]; and 63% in Hong Kong [47]. HCWs’ hesitancy mainly originated from concerns regarding the safety of the COVID-19 vaccines. A few combined interventions, including education and training sessions, easy vaccine accessibility, and rewards after vaccination, etc., might increase vaccination uptake [48]. Our study suggests that parental hesitancy toward category B vaccines might be associated with the HCWs’ hesitancy toward category B vaccines. Because category B vaccines are optional and billed, recommendations by HCWs have an important role in parents’ decisions, especially for first-time parents. Effective interventions need to be implemented to reduce HCWs’ hesitancy toward vaccines, as they may also indirectly affect the vaccine hesitancy of parents. Further evidence of the association between parental vaccine hesitancy and HCW vaccine hesitancy is needed.

This study had some limitations that need to be considered when interpreting the findings. First, the study population involved mainly young parents, with 83.1% aged <36 years. The participants were only recruited from vaccination clinics in Wuxi, and 60% of the children were aged <3 years. Selection bias may have limited the representativeness of the sample. Second, vaccine hesitancy and vaccine willingness are influenced by many complex factors, which might vary across different times, policies, and health systems. The willingness to obtain COVID-19 vaccination might vary. Third, in the actual survey, parents filled out the questionnaire while taking care of their children. If the child was crying, it may have affected the parents’ attention to the survey. Fourth, more influencing factors associated with vaccine hesitancy and vaccine willingness should be identified. Fifth, COVID-19 vaccinations for children are not yet approved. However, obtaining data on Chinese parents’ intentions toward COVID-19 vaccines could help inform the development of child-immunization policies in the future. Finally, the association between parental vaccine hesitancy and HCW vaccine hesitancy was assessed only in a small sample size. More efforts are needed in sampling-size selection and survey methods.

## 9. Conclusions

In China, parents are more hesitant about category B vaccines than category A vaccines. More than 40% of parents surveyed expressed hesitancy or refusal of the COVID-19 and influenza vaccinations. Parental hesitancy about category B vaccines was associated with parental willingness to receive a COVID-19 vaccine. Further, parental hesitancy toward category B vaccines was associated with HCW hesitancy toward category B vaccines. Effective interventions need to be implemented to address parental vaccine hesitancy and improve vaccine uptake.

## Figures and Tables

**Figure 1 vaccines-09-00342-f001:**
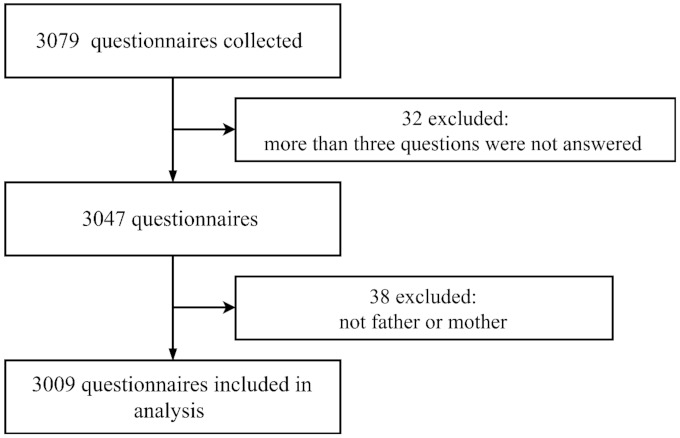
Flow chart for parent questionnaires.

**Figure 2 vaccines-09-00342-f002:**
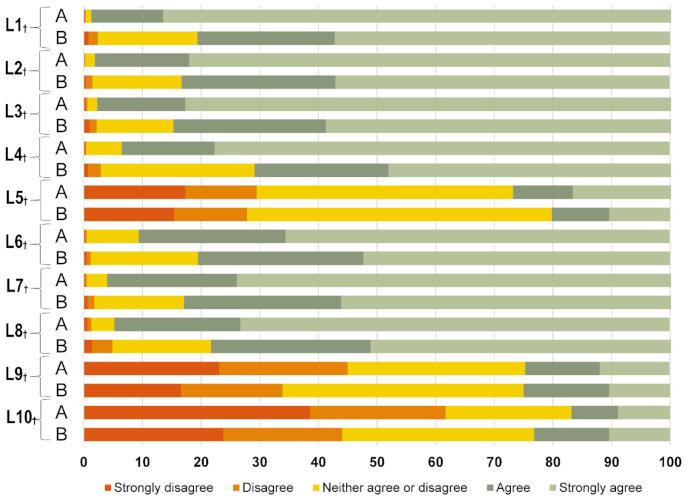
Distribution of parental responses to the 10 items on the vaccine-hesitancy scale. ^†^
*p* < 0.05. A = category A vaccine; B = category B vaccine.

**Figure 3 vaccines-09-00342-f003:**
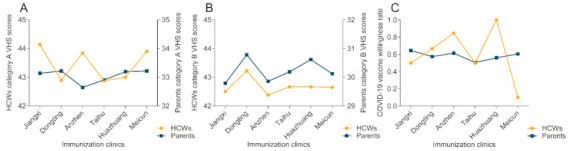
Relationship of vaccine hesitancy and willingness between parents and healthcare workers by immunization clinics. (**A**) Category A vaccine hesitancy between parents and healthcare workers by immunization clinics, *r* = 0.143, *p* = 0.787; (**B**): different category A vaccine hesitancy between parents and healthcare workers by immunization clinics, *r* = 0.928, *p* = 0.008; (**C**): COVID-19 vaccine willingness between parents and healthcare workers by immunization clinics, *r* = −0.174, *p* = 0.742. HCWS = healthcare workers.

**Table 1 vaccines-09-00342-t001:** The 10-item Likert scale.

No.	Content
L1	Childhood vaccines are important for my child’s health
L2	Childhood vaccines are effective
L3	Having my child vaccinated is important for the health of others in my community
L4	All childhood vaccines offered by the government program in my community are beneficial
L5	New vaccines carry more risks than older vaccines
L6	The information I receive about vaccines from the vaccine program is reliable and trustworthy
L7	Getting vaccines is a good way to protect my child/children from disease
L8	Generally I do what my doctor or healthcare provider recommends about vaccines for my child/children
L9	I am concerned about serious adverse effects of vaccines
L10	My child/children does/do not need vaccines for diseases that are not common anymore

**Table 2 vaccines-09-00342-t002:** Participants’ sociodemographic information.

Sociodemographics	Data, N (%)
Children	-
Age, year (Mean ± SD)	2.24 ± 2.47
Age group	-
<1	1067 (35.5)
1–<2	467 (15.5)
2–<3	321 (10.7)
3–<4	298 (9.9)
4–<5	313 (10.4)
5–<6	142 (4.7)
≥6	401 (13.3)
Gender	-
Male	1528 (50.8)
Female	1481 (49.2)
Firstborn	-
Yes	1982 (65.9)
No	1027 (34.1)
Parents	-
Relationship with child	-
Mother	2245 (74.6)
Father	764 (25.4)
Age, year(Mean ± SD)	31.36 ± 4.46
Age group	-
<26	249 (8.3)
26–<31	1091 (36.3)
31–<36	1162 (38.6)
36–<41	417 (13.9)
≥41	90 (3.0)
Educational level	-
Junior high school or below	349 (11.6)
High school graduate or equivalent	563 (18.7)
College or equivalent	1878 (62.4)
Master’s Diploma or above	219 (7.3)
Annual household income (RMB 10,000)	-
<5	213 (7.1)
5–<10	876 (29.1)
10–<15	780 (25.9)
≥15	1140 (37.9)
Healthcare occupation	-
Yes	186 (6.2)
No	2823 (93.8)
Healthcare workers	-
Age, year (Mean ± SD)	36.02 ± 9.44
Gender	-
Male	20 (23.3)
Female	66 (76.7)
Educational level	-
High school graduate or equivalent	5 (5.8)
College or equivalent	81 (94.2)
Specialty	-
Public health	27 (30.2)
Clinical medicine	10 (11.6)
Nursing	45 (52.3)
Others	5 (5.8)
Title	-
Junior	47 (53.5)
Intermediate	22 (25.6)
Senior	15 (17.4)
Others	3 (3.5)
Working years	-
<1	1 (1.2)
1–<5	19 (22.1)
5–<10	21 (24.4)
10–<15	13 (15.1)
≥15	32 (37.2)
Annual household income (RMB 10,000)	-
<5	6 (7.0)
5–<10	17 (19.8)
10–<15	24 (27.9)
≥15	39 (45.3)

USD 1 = RMB 6.8148. The currency-exchange data was retrieved from http://www.pbc.gov.cn/diaochatongjisi/resource/cms/2020/10/2020101616110773362.htm (accessed on 29 December 2020, Nanjing, Jiangsu, China).

**Table 3 vaccines-09-00342-t003:** Univariate and multivariate factors of parental category A and B vaccine hesitancy.

Characteristic	Category A VHS				Category B VHS			
	Univariate, *β* (95% CI)	*p* Value	Multivariate, *β* (95% CI)	*p* Value *	Univariate, *β* (95% CI)	*p* Value	Multivariate, *β* (95% CI)	*p* Value *
Children								
Age group								
<1	Reference	-	Reference	-	Reference	-	-	-
1 to <2	−0.002 (−0.013, 0.008)	0.647	-	-	0.005 (−0.010, 0.019)	0.542	-	-
2 to <3	0.000 (−0.012, 0.012)	0.992	-	-	−0.012 (−0.029, 0.005)	0.153	-	-
3 to <4	−0.021 (−0.034, −0.009)	0.001	−0.018 (−0.029, −0.006)	0.012	−0.010 (−0.028, 0.007)	0.250	-	-
4 to <5	−0.013 (−0.025, −0.001)	0.038	-	-	−0.003 (−0.020, 0.014)	0.718	-	-
5 to <6	−0.004 (−0.021, 0.013)	0.638	-	-	−0.002 (−0.026, 0.021)	0.845	-	-
≥6	−0.008 (−0.020, 0.003)	0.139	-	-	0.001 (−0.015, 0.017)	0.895	-	-
Gender								
Male	Reference	-	-	-	Reference	-	-	-
Female	−0.004 (−0.011, 0.003)	0.286	-	-	−0.003 (−0.013, 0.007)	0.530	-	-
Firstborn								
No	Reference	-	Reference	-	Reference	-	-	-
Yes	−0.009 (−0.016, −0.002)	0.018	−0.010 (−0.017, −0.003)	0.012	0.004 (−0.006, 0.014)	0.435	-	-
Parents								
Relationship with child								
Mother	Reference	-	Reference	-	Reference	-	Reference	-
Father	0.011 (0.003, 0.019)	0.009	0.011 (0.003, 0.019)	0.012	0.013 (0.002, 0.024)	0.023	0.013 (0.002, 0.024)	0.023
Age group								
<26	Reference	-	-	-	Reference	-	-	-
26 to <31	0.010 (−0.004, 0.023)	0.148	-	-	0.013 (−0.006, 0.031)	0.180	-	-
31 to <36	0.007 (−0.007, 0.020)	0.316	-	-	0.010 (−0.009, 0.028)	0.314	-	-
36 to <41	0.006 (−0.009, 0.022)	0.418	-	-	0.008 (−0.013, 0.029)	0.463	-	-
≥41	0.018 (−0.006, 0.041)	0.136	-	-	0.023 (−0.010, 0.056)	0.165	-	-
Educational level								
Junior high school	Reference	-	-	-	Reference	-	-	-
High school graduate or equivalent	−0.003 (−0.016, 0.010)	0.697	-	-	0.004 (−0.014, 0.023)	0.632	-	-
College or equivalent	0.004 (−0.007, 0.015)	0.470	-	-	0.013 (−0.003, 0.028)	0.108	-	-
Master’s Diploma or above	0.010 (−0.006, 0.027)	0.225	-	-	0.018 (−0.005, 0.041)	0.125	-	-
Annual household income (RMB 10,000)								
<5	Reference	-	Reference	-	Reference	Reference	-	-
5 to <10	0.017 (0.002, 0.031)	0.026	0.008 (0.000, 0.016)	0.042	0.019 (−0.001, 0.039)	0.068	-	-
10 to <15	0.009 (−0.005, 0.024)	0.209	-	-	0.011 (−0.009, 0.032)	0.275	-	-
≥15	0.009 (−0.005, 0.023)	0.225	-	-	0.019 (−0.001, 0.039)	0.066	-	-
Healthcare occupation								
Yes	Reference	-	-	-	Reference	-	-	-
No	−0.013 (−0.028, 0.001)	0.070	-	-	−0.012 (−0.032, 0.009)	0.263	-	-

* The *p* value was adjusted using the false discovery rate method.

**Table 4 vaccines-09-00342-t004:** Univariate and multivariate factors of parental COVID-19 and influenza vaccination acceptance, OR (95% CI) *.

Variables	COVID-19 Vaccination		Influenza Vaccination	
	Univariate	Multivariate	Univariate	Multivariate
Children				
Age group				
<1	1.177 (0.926, 1.497)	-	0.465 (0.367, 0.590)	0.121 (0.374, 0.602)
1–<2	1.033 (0.782, 1.364)	-	0.544 (0.414, 0.714)	0.140 (0.422, 0.731)
2–<3	0.998 (0.734, 1.357)	-	0.412 (0.305, 0.557)	0.154 (0.310, 0.568)
3–<4	1.005 (0.734, 1.376)	-	1.255 (0.912, 1.727)	0.164 (0.909, 1.726)
4–<5	1.062 (0.780, 1.448)	-	0.939 (0.691, 1.276)	0.157 (0.678, 1.255)
5–<6	0.879 (0.587, 1.317)	-	1.429 (0.940, 2.172)	0.215 (0.891, 2.067)
≥6	Reference		Reference	Reference
Gender				
Male	0.992 (0.857, 1.147)	-	0.874 (0.758, 1.009)	-
Female	Reference	-	Reference	-
Firstborn				
No	0.988 (0.847, 1.152)	-	0.873 (0.751, 1.015)	-
Yes	Reference	-	Reference	-
Parents				
Relationship with child				
Mother	1.024 (0.866, 1.210)	-	1.262 (1.070, 1.487)	-
Father	Reference	-	Reference	-
Age group				
<26	0.997 (0.608, 1.635)	-	0.370 (0.224, 0.611)	-
26–<31	1.045 (0.672, 1.624)	-	0.543 (0.347, 0.849)	-
31–<36	0.904 (0.583, 1.403)	-	0.713 (0.456, 1.114)	-
36–<41	0.753 (0.473, 1.199)	-	0.886 (0.551, 1.424)	-
≥41	Reference	-	Reference	-
Educational level				
Junior high school or below	1.563 (1.112, 2.197)	1.563 (1.112, 2.197)	0.586 (0.416, 0.824)	-
High school graduate or equivalent	2.139 (1.556, 2.939)	2.139 (1.556, 2.939)	0.732 (0.534, 1.003)	-
College or equivalent	1.477 (1.116, 1.955)	1.477 (1.116, 1.955)	0.909 (0.685, 1.206)	-
Master’s Diploma or above	Reference	Reference	Reference	-
Annual household income (RMB 10,000)				
<5	1.044 (0.777, 1.403)	-	0.619 (0.462, 0.831)	0.153 (0.504, 0.919)
5–<10	1.346 (1.124, 1.612)	-	0.662 (0.554, 0.790)	0.093 (0.602, 0.867)
10–<15	1.212 (1.007, 1.459)	-	0.688 (0.573, 0.827)	0.096 (0.607, 0.884)
≥15	Reference	-	Reference	Reference
Healthcare occupation				
No	0.927 (0.684, 1.257)	-	1.111 (0.826, 1.495)	-
Yes	Reference	-	Reference	-

* Yes vs. not sure and no.

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
