# Peer review of "Vaccine Hesitancy: COVID-19 and Influenza Vaccine Willingness among Parents in Wuxi, China—A Cross-Sectional Study"

_vaccines, 2021, doi:10.3390/vaccines9040342_

Round 1
Reviewer 1 Report
The manuscript “Vaccine hesitancy, COVID-19 and influenza vaccine willingness among parents in Wuxi, China: a cross-section study”, is a retrospective analysis of the vaccine hesitancy, including COVID-19 and influenza vaccination, performed in Wuxi, a region of Eastern China.
The data collected by the authors are very interesting and given the complexity involved, the author has produced many positive and welcome outcomes. Overall, this research is well written, and the content of this manuscript is of major interest. Nevertheless, the following issues need to be addressed:
Affiliations: Usually numbers instead of letters (as superindex) are used to indicate the authors affiliations.
Line 9: Why this affiliation is underlined?
The resolution of Figure 1 and, especially, of Figure 3 is very low. Please replace these figures. Also check the figure legend 3 and correct it.
Line 240: What is this link?
In table 3 and 4, what indicate the word “Ref”? What do you mean?ù
Line 281: Please correct this title
Funding, Contributors etc..: In these sections there are several typographical mistakes
Reference list: Please check the correct reference list style required by Vaccines
Author Response
The manuscript “Vaccine hesitancy, COVID-19 and influenza vaccine willingness among parents in Wuxi, China: a cross-section study”, is a retrospective analysis of the vaccine hesitancy, including COVID-19 and influenza vaccination, performed in Wuxi, a region of Eastern China.
The data collected by the authors are very interesting and given the complexity involved, the author has produced many positive and welcome outcomes. Overall, this research is well written, and the content of this manuscript is of major interest. Nevertheless, the following issues need to be addressed:
Point 1: Affiliations: Usually numbers instead of letters (as superindex) are used to indicate the authors affiliations.
Response 1: Thank you for the suggestion, and we have used numbers instead of letters.
Point 2: Line 9: Why this affiliation is underlined?
Response 2: It may be a typographical error and we have deleted the underscore.
Point 3: The resolution of Figure 1 and, especially, of Figure 3 is very low. Please replace these figures. Also check the figure legend 3 and correct it.
Response 3: Thank you for your suggestions. We replaced Figure 1 and 3 with more high resolution of figures (1200 DPI) in the manuscript. The figure legend 3 has been corrected (Line 330-Line335).
Point 4: Line 240: What is this link?
Response 4: Thank you for your suggestions. The link provided the data about RMB converting into US$. We have revised the sentence to make it clear (Line 250-251).
Point 5: In table 3 and 4, what indicate the word “Ref”? What do you mean?ù
Response 5: Thank you for your suggestions. The word “Ref “is the abbreviation of reference. We have replaced the “Ref” with “Reference” through the paper.
Point 6: Line 281: Please correct this title
Response 6: Thank you for your suggestions. We have revised it.
Point 7: Funding, Contributors etc..: In these sections there are several typographical mistakes
Response 7: Thank you for your suggestions. We have corrected these sections.
Point 8: Reference list: Please check the correct reference list style required by Vaccines
Response 8: We have corrected the reference list style according to requirements.
Reviewer 2 Report
I was invited to revise the paper entitled "Vaccine hesitancy, COVID-19 and influenza vaccine willingness among parents in Wuxi, China: a cross-section study". It aimed to evaluate vaccine hesitancy between two group, the parents’ willingness to acquire COVID-19 and influenza vaccines for their children and the association between vaccine hesitancy among parents and HCWs. This is an interesting paper that improve the knowledge in this topic during this epidemic.
Introduction well describe the stdy background. Methods are clear and results are well presented.
I have some observation that need to be addressed:
- Validation of the questionnaire need to be reported;
- Multivariate analyses need to be corrected for multiple comparison, such as False Discovery Rate;
- Linear regression analyses is not adequate for not-normally distributed variables;
- About discussion section, HCWs vaccine hesitancy should be compared with other studies such as 10.3390/vaccines8020248
Author Response
I was invited to revise the paper entitled "Vaccine hesitancy, COVID-19 and influenza vaccine willingness among parents in Wuxi, China: a cross-section study". It aimed to evaluate vaccine hesitancy between two group, the parents’ willingness to acquire COVID-19 and influenza vaccines for their children and the association between vaccine hesitancy among parents and HCWs. This is an interesting paper that improve the knowledge in this topic during this epidemic.
Introduction well describe the stdy background. Methods are clear and results are well presented.
I have some observation that need to be addressed:
Point 1: Validation of the questionnaire need to be reported;
Response 1: Thank you for the suggestion. We have provided the results of validation of the questionnaire in the manuscript and supplement materials. We performed an exploratory factor analysis and confirmatory factor analysis to assess reliability and validity of vaccine hesitancy scale. The results suggested that reliability and validity of 10-item VHS were acceptable.
Point 2: Multivariate analyses need to be corrected for multiple comparison, such as False Discovery Rate;
Response 2: Thank you for the suggestion. We have used the false discovery rate correction (FDR) to control the family-wise type I error rate and reported FDR adjusted p-value.
Point 3: Linear regression analyses is not adequate for not-normally distributed variables;
Response 3: Thank you for the suggestion. We analysed the data using generalized linear model (GLM) instead of linear regression model. This method was used in the previous study to explore the association between mother’s education level and vaccine hesitancy scale scores [1]. GLM is useful tool for analyzing nonnormal data [2]. It handles nonnormal data by using link functions and exponential family.
[1] Wagner AL, Masters NB, Domek GJ, et al. Comparisons of Vaccine Hesitancy across Five Low- and Middle-Income Countries. Vaccines (Basel) 2019;7(4):155.
[2] Bolker BM, Brooks ME, Clark CJ, et al. Generalized linear mixed models: a practical guide for ecology and evolution. Trends Ecol Evol 2009;24(3):127-135.
Point 4: About discussion section, HCWs vaccine hesitancy should be compared with other studies such as 10.3390/vaccines8020248
Response 4: Thank you for the suggestion. We have added some contents about HCWs vaccine hesitancy in discussion section.
Round 2
Reviewer 2 Report
Authors addressed all point. The paper is now acceptable for publication.